# Impact of SOCE Abolition by ORAI1 Knockout on the Proliferation, Adhesion, and Migration of HEK-293 Cells

**DOI:** 10.3390/cells10113016

**Published:** 2021-11-04

**Authors:** Alexandre Bokhobza, Nathalie Ziental-Gelus, Laurent Allart, Oksana Iamshanova, Fabien Vanden Abeele

**Affiliations:** Inserm U1003, Laboratory of Cell Physiology, Université de Lille, 59650 Villeneuve d’Ascq, France; nathalie.ziental-gelus@univ-lille.fr (N.Z.-G.); laurent.allart@inserm.fr (L.A.); iamshanova.oksana@gmail.com (O.I.)

**Keywords:** calcium, ORAI protein, store-operated calcium entry, proliferation, migration, adhesion, HEK-293

## Abstract

Store-operated calcium entry (SOCE) provided through channels formed by ORAI proteins is a major regulator of several cellular processes. In immune cells, it controls fundamental processes such as proliferation, cell adhesion, and migration, while in cancer, SOCE and *ORAI1* gene expression are dysregulated and lead to abnormal migration and/or cell proliferation. In the present study, we used the CRISPR/Cas9 technique to delete the *ORAI1* gene and to identify its role in proliferative and migrative properties of the model cell line HEK-293. We showed that ORAI1 deletion greatly reduced SOCE. Thereby, we found that this decrease and the absence of ORAI1 protein did not affect HEK-293 proliferation. In addition, we determined that ORAI1 suppression did not affect adhesive properties but had a limited impact on HEK-293 migration. Overall, we showed that ORAI1 and SOCE are largely dispensable for cellular proliferation, migration, and cellular adhesion of HEK-293 cells. Thus, despite its importance in providing Ca^2+^ entry in non-excitable cells, our results indicate that the lack of SOCE does not deeply impact HEK-293 cells. This finding suggests the existence of compensatory mechanism enabling the maintenance of their physiological function.

## 1. Introduction

Calcium (Ca^2+^) is a major intracellular second messenger controlling several fundamental cellular processes [1]. Therefore, Ca^2+^ signals are finely regulated both spatially and temporally due to a variety of specialized Ca^2+^ channels, exchangers, and transporters [2]. This fine regulation results in a tight control of crucial cellular functions such as proliferation and migration [3]. It is widely accepted that one of the main Ca^2+^ entry pathways in non-excitable cells is represented by store-operated calcium entry (SOCE) [4]. SOCE is defined by Ca^2+^ entry across the cell plasma membrane (PM) consecutive to Ca^2+^ depletion from the endoplasmic reticulum (ER) [5]. The molecular components of SOCE are represented by the ER-resident protein family stromal interaction molecule (STIM), including STIM1 and STIM2, and the plasma membrane protein ORAI family, including ORAI1, ORAI2, and ORAI3. Specifically, the ER-residing STIM1 protein bears an EF-hand motif facing the ER lumen that detects variations in the ER Ca^2+^ concentration [6]. Following a drop in the ER Ca^2+^ content, the STIM1 protein undergoes a conformational change leading to its oligomerization and translocation to the ER–PM junction, where it binds the ORAI1 protein and triggers Ca^2+^ influx [7]. Since the identification of ORAI1 as the main effector of SOCE, its functional importance in (patho) physiological processes has been highlighted extensively. For example, patients bearing *ORAI1* loss-of-function mutations were reported to develop severe conditions such as muscular hypotonia, ectodermal dysplasia, and severe combined immunodeficiency (SCID)-like disease [8]. In addition, the proliferation of endothelial cells is inhibited through cell-cycle arrest after short-interfering RNA (siRNA) treatment against *ORAI1* [9]. Similarly, the artificial downregulation of ORAI1 expression impairs the differentiation of myoblasts [10]. Moreover, ORAI1 knockdown inhibits the migration of myeloblastic cells [11]. Finally, in cancer, ORAI1 and SOCE are significantly dysregulated [12]. Nonetheless, previous studies concluded on opposites roles for ORAI1 and SOCE on cell proliferation features depending on the cellular model studied. On the one hand, siRNA treatment against ORAI1 as well as pharmacological SOCE blockers induce cell-cycle arrest in different types of cancer [13]. On the other hand, the pharmacological and genetic inhibition of SOCE has no effect on the proliferation of primary cultures of human metastatic renal cellular carcinoma [14]. Finally, in a physiological context, the absolute requirement of ORAI1 and SOCE in the control of fundamental cellular processes remains questionable. For example, a publication from our laboratory suggests that ORAI1 downregulation is responsible for a decrease in HEK-293 proliferation [15]. In a consecutive study, Borowiec et al. [16] added that ORAI proteins are important for the growth of HEK-293 cell but SOCE is dispensable. Similar differences regarding the role of SOCE and ORAI1 have been observed in cell migration properties. On the one hand, ORAI1 downregulation impairs the migration of breast cancer cells [17] and decreases the invadopodia formation of melanoma cells [18]. On the other hand, Zuccolo et al. [19] showed that ORAI1 and SOCE are not involved in the migration of primary cells issued from colorectal carcinoma, thus questioning the physiological role of ORAI1 and SOCE during the migration process. Interestingly, the roles of SOCE and ORAI1 in cellular physiology have mostly been studied by downregulating *ORAI1* gene expression or by modulating SOCE functionality with pharmacological agents. Although these techniques have proven to be powerful, pitfalls such as residual endogenous ORAI1 protein after knockdown, lack of specificity of pharmacological agents inhibiting SOCE, and potential compensation due to the expression of other ORAI isoforms might prevent the identification of clear roles for ORAI1 and SOCE in cell physiology. In addition, studies on physiological roles of ORAI1 have mostly been performed in immune cells or in the (patho) physiological context of cancer. By contrast, in HEK-293 cells, SOCE has mostly been studied to understand the molecular mechanisms of its activation. However, its fundamental role in the maintenance of HEK-293 physiology has not been addressed. Thus, in this study, we decided to take advantage of the advent of the CRISPR/Cas9 technique—allowing genome editing—to generate complete ORAI1 knockout (KO) in HEK-293 cells to assess the involvement of the ORAI1 protein and SOCE in proliferation and migration.

## 2. Material and Methods

### 2.1. Cell Culture

HEK-293 cells were obtained from the European Collection of Authenticated Cell Cultures (ECACC) through Sigma-Aldrich (Sigma-Aldrich, St. Quentin Fallavier, France). Cells were maintained in high-glucose, GlutaMAX™ Dulbecco’s modified Eagle medium (DMEM; Thermo Fischer Scientific, Illkirch, France) supplemented with 10% heat-inactivated fetal bovine serum (FBS; Dominique Dutscher, Brumath, France) and cultured in 75 cm² flasks kept at 37 °C in a humidified incubator containing 5% CO_2_. All the cells used in this study were validated as being mycoplasma free by using Hoechst staining.

### 2.2. Transfections

PEI Max 40K (Biovalley, Nanterrre, France) was used to perform all transfections. PEI Max solution was prepared according to the manufacturer’s instructions. In brief, 100 mg of PEI Max 40K was dissolved in 90 mL of double-distilled water (ddH_2_O), with the pH adjusted to 7 by using NaOH before adjusting the volume to 100 mL to obtain a stock solution at a concentration of 1 mg/mL. The solution was sterilized by filtration through a 0.22 µm pore-size filter, and the stock solution was aliquoted in several smaller volumes and kept at 4 °C for further use.

For transfections, HEK-293 cells were grown in FBS-containing medium in 60 mm dishes until they reached 70% confluency. Transfection solution was prepared by mixing plasmid DNA and PEI Max at a concentration of 1 µg of DNA for 2 µL of PEI Max solution (1 µg/µL) in FBS-free DMEM. Following gentle flicking of the tube and an incubation period of 15 min, the cell medium was discarded and replaced with fresh FBS-containing medium. Subsequently, transfection solution was added on top of the cells in a dropwise manner. Subsequent experiments were carried out 48 h after transfections.

### 2.3. CRISPR/Cas9 KO Generation

Guide RNA (gRNA) was designed by using the CRISPOR website [20]. Cutting efficiencies of six gRNAs (three gRNAs for each desired localization) were assessed by using the T7 Endonuclease I assay (New England Biolabs, Evry, France) according to the manufacturer’s instructions. The choice of gRNAs was dictated by their efficiency and the number of predicted off-target cutting sites. The gRNA sequences are shown in Table 1. Of note, off-target analysis was performed using HEK-293 genome database (from: http://HEK-293genome.org/v2/, access date: 25 October 2021). gRNA1 possesses 49 potential off-targets sites (with a mismatch number ≥ 4), with 6 sites located in exonic region (excluding the *ORAI2* and *ORAI3* genes), and gRNA2 possesses 108 potential off-targets sites (with a mismatch number ≥ 3), with 10 sites located in exonic region (including one off-target site in the *ORAI3* gene. Of note, one of the mismatches targeting *ORAI3* gene is located on the first nucleotide preceding the protospacer adjacent motif (PAM) which greatly reduces the probability of double strand break (DSB) induction [21]). CRISPR/Cas9 plasmids were purchased from Addgene: pSpCa9(BB)-2A-GFP (PX458). gRNA1 and gRNA2 were cloned into the plasmids according to the Zhang lab protocol [22]. HEK-293 cells were transfected with gRNA1 and gRNA2-containing plasmids as described above. Two days after transfection, cells were detached, counted, and diluted to obtain a cell concentration of approximately 0.6 cell/100 µL of medium. One hundred microliters of cell suspension was added into a 96-well plate prefilled with 100 µL of warm FBS-containing DMEM. Cells were left to grow for approximately 2–3 weeks until a noticeable change in the media color, indicating the presence of a growing single-cell colony. Single-cell colonies were further amplified, and screening for successful genomic editing was performed.

### 2.4. Genomic DNA (gDNA) Extraction

gDNA was extracted by using the QuickExtract DNA extraction solution (Euromedex, Souffelweyersheim, France). In brief, an appropriate volume of DNA extraction solution was added to the cell pellet, and the tubes were vortexed for 30 s before a 6 min incubation at 65 °C. Subsequently, the tubes were vortexed for 30 s and incubated at 96 °C for 2 min. Following the cooling of the tubes, a last vortex step was performed. The gDNA concentration was determined by using a spectrophotometer. gDNA was stored at −20 °C or directly used for polymerase chain reaction (PCR).

### 2.5. PCR and Genomic Sequencing

PCR using gDNA was performed with the OneTaq Hot Start 2X MasterMix GC buffer polymerase (New England Biolabs, Evry, France) or Q5 Hot Start 2X MasterMix polymerase (New England Biolabs, Evry, France) to screen for genetic deletion and to sequence the *ORAI1* gene, respectively. Primer sequences used for PCR are indicated in Table 1. The starting gDNA quantity was 10 ng. PCR products were cloned for sequencing by using the NEB PCR cloning kit according to the manufacturer’s instructions. Plasmid sequencing was performed by Eurofins.

### 2.6. Quantitative Reverse Transcription PCR (RT-qPCR)

RT-qPCR of messenger RNA (mRNA) extracted from HEK-293 cells was performed by using the SsoFast Eva Green Supermix (Bio-Rad Laboratories, Cressier, France). RT-qPCR was performed on a CFX C1000 instrument (Bio-Rad Laboratories). Primer sequences are indicated in Table 1. Each experiment was repeated four times. RT-qPCR data were analyzed with the common base method, which accounts for the specific qPCR efficiency of each analyzed gene [23]. Glyceraldehyde-3-phosphate dehydrogenase (GAPDH) and TATA-box-binding protein (TBP) were used as housekeeping genes.

### 2.7. Immunobloting

Sodium dodecyl sulfate–polyacrylamide gel electrophoresis (SDS-PAGE) was performed by loading 50 µg of total protein lysate (without nuclei) extracted from HEK-293 cells on 12% polyacrylamide gels. Following electrophoresis, separated proteins were transferred to methanol-activated polyvinylidene fluoride (PVDF) membranes with the Pierce G2 Fast Blotter (Thermo Fischer Scientific). Membranes were incubated in 5% nonfat milk in TBS-T buffer (15 mM Tris-HCl, 140 mM NaCl, 0.05% Tween20^®^, pH 7.4) for 1 h at room temperature to block nonspecific protein binding. Subsequently, the membranes were incubated with primary antibodies overnight at 4 °C. The primary antibodies were anti-ORAI1 (Alomone, Jerusalemn, Israel: Acc-060), used at a 1/200 dilution, and anti-β-actin Sigma-Aldrich, St. Quentin Fallavier, France: A5441), used at a 1/5000 dilution. Following washes in TBS-T, the membranes were incubated for 1 h with horseradish peroxidase (HRP)-conjugated anti-mouse or anti-rabbit secondary antibodies diluted at 1/100,000 (Chemicon). Chemiluminescent detection of bound secondary antibodies was captured with Amersham Imager 600 (GE Healthcare Life Sciences) using the SuperSignal™ West Femto Maximum Sensitivity Substrate and the SuperSignal™ West Dura Maximum Sensitivity Substrate (Thermo Fischer Scientific, Illkirch, France).

### 2.8. Calcium Imaging

Experiments were carried out in modified Hank’s balanced salt solution (HBSS) containing:-For the 0 mM Ca^2+^ HBSS: NaCl 150 mM, KCl 5 mM, MgCl_2_ 3 mM, CaCl_2_ 0 mM, D-glucose 10 mM, HEPES 10 mM, EGTA 0.1 mM; pH 7.4 with NaOH;-For the 2 mM Ca^2+^ HBSS: NaCl 150 mM, KCl 5 mM, MgCl_2_ 1 mM, CaCl_2_ 2 mM, D-glucose 10 mM, HEPES 10 mM; pH 7.4 with NaOH;-For the 8 mM Ca^2+^ HBSS: NaCl 150 mM, KCl 5 mM, MgCl_2_ 0 mM, CaCl_2_ 8 mM, D-glucose 10 mM, HEPES 10 mM; pH 7.4 with NaOH;

Cells were plated 3 days before experiments on glass-bottom dishes precoated with poly-L-Lysine (Ibidi). The cytosolic Ca^2+^ concentration was measured by using the ratiometric dye Fura2-AM (Interchim) diluted in 2 mM Ca^2+^ HBSS solution to a final concentration of 1 µM. Cells were incubated in this solution for 45 min at 37 °C.

Following incubation, Fura2-AM solution was washed out twice with the 0 mM Ca^2+^ solution. Experiments were performed in 1 mL of 0 mM Ca^2+^ HBSS. At the desired time point, 100 µL of 0 mM Ca^2+^ HBSS containing 10 µM of thapsigargin (TG) was added (final TG concentration = 1 µM). Subsequently, 1 mL of 8 mM Ca^2+^ HBSS was added upon the previous solution into the dish. The free final Ca^2+^ concentration was 3.95 mM as calculated when using the online tool webmaxc standard (https://somapp.ucdmc.ucdavis.edu/pharmacology/bers/maxchelator/webmaxc/webmaxcS.htm, Accessed date: 20 October 2021).

Acquisitions were performed using a 20× objective lens on a Superfluor Nikon Eclipse Ti-series inverted microscope equipped with a 510/84 emission filter (wavelength/bandwidth) and coupled to an Rolera EM-C2 (Qimaging). Fura-2 was excited with a xenon lamp light (300 W), part of a DG4 illumination system (Sutter) equipped with the following filter pairs (wavelength/bandwidth): 340/26 and 387/11 nm. The acquisition software was Metafluor; data were then exported to Microsoft Excel. Time-series graphics were generated by using the PlotTwist webapp [24] for each individual experiment. Integral measurements were performed with Microsoft Excel, and data of three independent experiments were plotted by using the SuperPlotsOfData webapp [25].

### 2.9. Cell Cycle

For cell-cycle analysis, 1 × 10^5^ cells were seeded in individual wells of a 6-well plate. Three days post seeding, cells were trypsinized and then fixed with 70% ice-cold ethanol bath and pelleted by centrifugation at 200 g for 5 min. After three washes in phosphate-buffered saline (PBS), cells were resuspended in PBS containing 500 µg/mL of RNAse A. Cells were stained with propidium iodide (5 µg/mL) for 45 min in the dark, and then propidium iodide fluorescence was read by using a CyAn™ ADP Analyzer FACScan flow cytometer (Becton-Dickinson). At least 10,000 events per condition were used to interpret the data with Summit 4.3 (DAKO). A histogram was realized with Microsoft Excel, representing the data of six individual experiments.

### 2.10. Direct Cell Counting

For direct cell counting, 5 × 10^4^ cells were seeded into each well of a 6-well plate. Three days after seeding, cells from the first well were detached and counted; this procedure was repeated every 24 h until day 5 or 6 post seeding. The cell doubling time was calculated according to the following formula:$$Doubling\;time\;\left(hours\right)=\frac{Time\;between\;measurements\;\left(hours\right)\times \log2}{\log\left(Cell\;number\;\left[Day\;x\right]\right)-\log\left(Cell\;number\;\left[Day\;x-1\right]\right)}$$

The doubling time was calculated for each time point (i.e., 24, 48, 72, and 96 h). Data were plotted by using the SuperPlotsOfData webapp; the plots represent seven independent experiments.

### 2.11. Sulforhodamine B (SRB) Assay

For this assay, 5 × 10^3^ cells per well were plated in 96-well plates. Five days after seeding, the cell media was gently aspirated without touching adherent cellular layer and fixed with 50% trichloroacetic acid solution (Sigma-Aldrich, St. Quentin Fallavier, France) for 1 h at 4 °C. Following three washing steps with ddH_2_O, cells were stained with a solution of 0.4% sulforhodamine B (Sigma-Aldrich, St. Quentin Fallavier, France) in 1% acetic acid. Excess dye was removed by three washes with 1% acetic acid, and protein-bound dye was then dissolved in 10 mM Tris solution (pH 10.5). Absorbance was measured at 510 nm with a TriStar2 Multimodal Reader LB942 (Berthold Technologies). Data were plotted by using the SuperPlotsOfData webapp; the plots represent three independent experiments.

### 2.12. Adhesion Assay

Adhesion assays were performed in fibronectin precoated 96-well plates. In brief, 1 mg of lyophilized fibronectin (Sigma-Aldrich, St. Quentin Fallavier, France, F2006) was reconstituted in 1 mL of ddH_2_O. This solution was further diluted in sterile PBS (Thermo Fischer Scientific, Illkirch, France, 10010-015) to a concentration of 10 µg/mL, and then, 20 µL was distributed into 96-well plate. Plates were then left for 5 min at room temperature and, after aspiration of fibronectin solution, left uncovered under the hood for 30 min to air-dry. Next, 1 × 10^3^ cells were subsequently loaded in each well and incubated at 37 °C for 1 h. Then, the medium was removed, and the wells were washed once with PBS before fixation with 70% ethanol for 20 min at 4 °C. Following two washes with PBS, cell nuclei were stained for 15 min at room temperature using Hoechst solution (2.5 µg/mL). Pictures of the whole well surface were taken using the 4× objective lens of a Nikon Eclipse Ti-E microscope. Cell nuclei were counted automatically with an ImageJ macro. Data were plotted by using the SuperPlotsOfData webapp; for each individual experiment, 6–8 wells were used, and the experiments were repeated 4 times.

### 2.13. Wound Healing Assay

To ensure a constant wound size within each experiment, a wound healing experiment was performed by using a cell culture insert from Ibidi. An insert delimiting a 500 µm (±100 µm) cell-free space between compartments was placed in each well of a 12-well plate. Next, 70 µL of DMEM supplemented with 10% FBS and containing 9 × 10^4^ cells was added to each compartment of the insert. One milliliter of DMEM supplemented with 10% FBS was added outside of the insert. Cells were left to attach for 48 h before removal of the insert, creating a cell-free gap between the two compartments. Following insert removal, 12-well plates were placed on the automated stage of a Nikon Eclipse Ti-E microscope equipped with a chamber to maintain a temperature at 37 °C and 5% CO_2_, and pictures of the center of the wound were taken with the 4× objective lens. The position of the initial pictures was saved for each well to acquire the same field of each well automatically every hour for a period of 48 h. Wound closure was analyzed by using an ImageJ macro modified from the BioImage Informatics Index (biii.eu). Time-series graphs were generated by using the PlotTwist webapp. Wound closure data, ratio of the half closure time to the full closure, and healing speed data were plotted by using the SuperPlotsOfData webapp. For each individual experiment, three wounds were examined, and each experiment was repeated five times.

### 2.14. Boyden Chamber Migration Assay

Boyden migration assays were carried out by using a Transwell^®^ insert with a pore size of 8 µm (Corning Life Sciences, Amsterdam, Netherlands). The upper part of the Transwell^®^ insert was filled with 500 µL of DMEM containing 2 × 10^5^ cells; the Transwell^®^ insert was then immersed in a well containing 500 µL of DMEM supplemented with 10% FBS. After 24 h, the inserts were removed; cells from the upper part were eliminated by scraping, while cells located on the lower part were fixed using ice-cold methanol. Following two washing steps with ddH_2_O, cells were stained with crystal violet (1% crystal violet solution in 20% methanol). To assess the number of cells that migrated, five randomly selected fields of the insert were captured by using the 10× objective lens of a Nikon Eclipse TS100 microscope. Cells were counted manually by using ImageJ software. For each condition, experiments were repeated six times with one Transwell^®^ used per assay. Data were plotted by using the SuperPlotsOfData webapp.

### 2.15. Data Representation and Statistical Analysis

To provide the reader with the maximum amount of information about the data obtained, SuperPlots were generated by using the SuperPlotsOfData webapp whenever possible. The key feature of the SuperPlots is the fact that every single data point is represented on the plot, while statistical representation uses only the mean or median value of each single experiment [26]. Similarly, continuous data (time series) were generated by using the PlotTwist webapp because it represents every single trace per condition as well as their mean or median values.

For all experiments and within technical replicates, median values were used whenever the distribution of the data was considered non-normal (evaluated with the Shapiro–Wilk test, *p* < 0.05); otherwise, mean values were used. For statistical analysis, the mean of biological replicates was used. Error bars represent the 95% confidence interval (CI), except when stated otherwise. Of note, the 95% CI represents the interval in which the real value for a specific experiment is included, thus giving a visual interpretation of the significance of a result. The 95% CI was calculated from the mean of biological replicates [27]. Statistical analysis was performed by using the differences between the means of the conditions, except if stated otherwise. Because the compared cell lines were modified genetically and evolved individually in the time between the KO generation and when the experiments were performed, we considered them a separate, individual cell line. Thus, we considered the data to be unpaired, and, consequently, we used Welch’s *t*-test for statistical analysis, except if stated otherwise.

## 3. Results

### 3.1. Generation of ORAI1 KO in HEK-293 Cells

The *ORAI1* gene comprises two exons and is located on chromosome 12. *ORAI1* mRNA can be translated into two ORAI1 isoforms: the full-length ORAI1 and a shorter version named ORAI1β (Figure 1A) [28]. Both isoforms possess four transmembrane domains as well as two interaction domains with the Ca^2+^ sensing protein STIM1. These domains, located at the N and C terminal tails of ORAI1, are necessary to ensure the normal function of the ORAI1 protein as an ion channel (Figure 1B) [29]. Therefore, to eliminate both isoforms and to generate complete ORAI1 KO, we used the CRISPR/Cas9 technique. In brief, the Cas9 protein cleaves DNA to create a (DSB) at the specific DNA location indicated by the guide RNA (gRNA). Because DNA repair mechanisms are error prone, they often result in a short insertion or deletion (InDel) that may disrupt the mRNA open-reading frame (ORF) and thus lead to the creation of protein KO. One of the limitations of this technique lies in the screening procedure enabling the identification of effective KO. To facilitate and ensure the generation of total ORAI1 KO, we decided to induce two DSB: one located before the classic *ORAI1* start codon (position –11 from the *ORAI1* coding sequence [CDS]) and the second located after the alternative *ORAI1* β start codon (position +251 from the *ORAI1* CDS). This double-gRNA approach has been validated by multiple teams for its ability to efficiently induce the deletion of the region situated between both gRNAs [30,31,32]. Thus, we hypothesized that our strategy induces a genomic deletion that includes the *ORAI1* start codon of the two identified isoforms and leads to the generation of complete ORAI1 KO (Figure 1C). We calculated that following the transfection of both gRNAs in our cells, a successful excision would result in a 263 bp deletion in *ORAI1*. To validate our strategy, we performed conventional PCR on gDNA using primers surrounding the cutting sites of each gRNA. Accordingly, wild-type HEK-293 cells showed a single band at the expected size of 647 bp, whereas cells transfected with gRNAs presented two distinct bands around 647 and 384 bp (Figure 2A). The larger band of 647 bp corresponds to cells that did not undergo genomic excision (*e.g*., due to unsuccessful transfection or unsuccessful DSB induction), and the lower band of 384 bp is consistent with a successful gDNA fragment excision. To isolate single-cell clones with successful gDNA fragment excision, we performed limiting dilution followed by PCR on gDNA. Single-cell-derived clones presenting multiple PCR bands (indicating a potential heterozygous genotype, clones 4 and 5) or bands of non-expected size (indicating unexpected genomic rearrangement, clones 2 and 7) were discarded (Figure 2B). Clones presenting a single band at the expected size for successful *ORAI1* gene deletion were selected (clones 1, 3, and 5; Figure 2B). Subsequently, their *ORAI1* gene was cloned and sequenced to confirm the successful deletion of the start codon. After clone expansion, effective KO generation was again validated by PCR (Figure 2C) and confirmed by immunoblot analysis (Figure 2D).

### 3.2. Effect of ORAI1 KO on SOCE

Because the main function of the ORAI1-formed channel is to provide Ca^2+^ entry after depletion of ER Ca^2+^ stores, we measured the level of SOCE in ORAI1 KO HEK-293 cells. TG is an irreversible inhibitor of sarco-/endo-plasmic reticulum Ca^2+^-ATPase (SERCA). Thus, ER Ca^2+^ store depletion was induced by applying TG in Ca^2+^-free medium (Figure 3A, grey area). Subsequent extracellular application of 4 mM Ca^2+^ led to plasmalemmal Ca^2+^ influx through ORAI channel (Figure 3A, yellow area). As expected, WT HEK-293 cells developed classic SOCE, while in ORAI1 KO cells, SOCE was impaired strongly. Thus, we confirmed that ORAI1 is the main effector of SOCE in HEK-293 cells (Figure 3A). For quantification of SOCE, the area under the curve following the Ca^2+^ addition (*i.e*., from 1200 to 2000 s) was integrated (Figure 3B). According to our results, SOCE was decreased by around 80% in ORAI1 KO compared with WT HEK-293 cells, but it was not totally abolished (Figure 3B).

### 3.3. Effect of ORAI1 KO on the Expression of ORAI2 and ORAI3

The three ORAI proteins (ORAI1, ORAI2, and ORAI3) are able to generate Ca^2+^ entry [33]. In addition to ORAI1, HEK-293 cells also endogenously express ORAI2 and ORAI3. Thus, the remaining SOCE observed in ORAI1 KO cells could be attributed to Ca^2+^ influx through ORAI2- and/or ORAI3-formed plasmalemmal channels. To check whether the deletion of ORAI1 led to compensatory overexpression of ORAI2 and ORAI3, we compared their mRNA expression levels in WT and ORAI1 KO HEK-293 cells. Our results indicate that both *ORAI2* and *ORAI3* are downregulated (Figure 4). Interestingly, two publications using triple KO in HEK-293 (*ORAI1*, *2*, and *3*) showed that after overexpression of ORAI2 and ORAI3 SOCE was restored in small proportion [34,35]. These results, together with the fact that ORAI2 and ORAI3 could form homomeric channels only in an overexpression system [33] (native currents provided through channels formed exclusively from endogenous ORAI2 or ORAI3 proteins were not reported yet), suggest that the residual Ca^2+^ entry observed in our cells is unlikely due to overexpression of ORAI2 or ORAI3. This is corroborated by the results from Yoast et al. [35], who performed patch-clamp experiment in ORAI1 KO HEK-293 cells and concluded on the absence of endogenous ORAI-provided current.

### 3.4. Effect of ORAI1 KO on the Proliferation of HEK-293 Cells

Because ORAI1 KO decreased SOCE by 80% in HEK-293 cells, we decided to study cellular processes known to be controlled by the intracellular Ca^2+^ concentration. We performed several experiments to elucidate the role of ORAI1 in proliferative properties. First, we calculated the doubling time by performing manual and automatic cell counting every 24 h for 5–6 consecutive days. There were no significant differences in doubling time between WT HEK-293 cells (25.77 h) and ORAI1 KO HEK-293 cells (26.81 h) (Figure 5A). Similarly, the normalized proliferation rate established by SRB staining after 5 days was not different when comparing WT HEK-293 cells (9.29) cells with ORAI1 KO HEK-293 cells (9.58) (Figure 5B). Finally, we performed cell-cycle analysis by assessing the cellular DNA content with propidium iodide staining. We established that the proportion of cells in the G_0_/G_1_, S, and G_2_/M phases was not significantly different between WT and ORAI1 KO HEK-293 cells: 58.07% versus 59.33% for the G_0_/G1 phase, 29.20% versus 28.30% for the S phase, and 12.73% versus 12.36% for the G_2_/M phase (Figure 5C). Overall, our data indicate that ORAI1 does not control the proliferation of HEK-293 cells.

### 3.5. Effect of ORAI1 KO on the Migration of HEK-293 Cells

Cell migration is a multistep process. Because the initiation of migration involves the release of cell adhesion spots from the substrate, we first focused on the adhesive properties. Our data indicate that WT and ORAI1 KO HEK-293 cells bear the same adhesive properties (Figure 6A). Further cell migration properties were investigated by a wound-healing assay. In brief, following the wound creation, the total wound area was measured every hour until full wound closure. Consequently, we evaluated the collective cell migration speed between WT and ORAI KO HEK-293 cells (Figure 6B). Interestingly, our results indicate that the collective cell migration of ORAI1 KO HEK-293 cells was faster during the first 10 h. From the tenth hour until complete wound closure, there were no differences in the collective migration speed between the two cell lines (Figure 6B). A comparison of the full wound closure time between the two cell lines did not show any differences (Figure 6C, left). Given that the initial wound size is susceptible to variation and because data on collective cell migration demonstrated the existence of differences in the collective migration speed of our cell lines, we decided to calculate the ratio between the half and full wound closure time. The result of this analysis confirmed that, at least until half wound closure, the collective migration of WT HEK-293 cells was significantly slower than the one for ORAI1 KO HEK-293 cells (Figure 6C, right). In addition to collective cell migration, we calculated the leading-edge front migration speed by measuring the shortest distance between the two migration fronts every 5 h (Figure 6D). The leading-edge migration of ORAI1 KO HEK-293 cells was faster than WT HEK-293 cells at hours 1–5 and 5–10, but there was no difference at hours 10–15. It is interesting to note that the speed of the leading-edge migration front increased throughout the experiment for both cell lines, and the final speed was the same. Taken together, our wound-healing data indicate that collective WT HEK-293 cell migration initiation is delayed compared with ORAI1 KO HEK-293 cells. To elucidate whether the effect observed is restricted to collective migration or whether it affects other properties of the migratory process, we conducted a chemotactic single-cell migration Transwell^®^ assay. After 24 h, the number of cells able to migrate on the lower side of the Transwell^®^ insert was not significantly different between WT HEK-293 cells (211 cells migrated on average) and ORAI1 KO HEK-293 cells (208 cells migrated on average) (Figure 6E). Therefore, the effect observed with our wound-healing experiment is restricted to collective migration properties.

## 4. Discussion

In this study, by using the CRISPR/Cas9 technique we created complete ORAI1 KO in HEK-293 cells. According to our results, ORAI1 is responsible for 80% of SOCE and, hence, is the main SOCE effector in HEK-293 cells, a finding consistent with previous reports [35,36,37]. Recent studies that generated ORAI1 KO in HEK-293 cells reported different remaining SOCE levels. Specifically, Cai et al. [36] observed that upon ORAI1 KO, SOCE was totally abolished. Similarly, Yoast et al. [35] reported nearly complete SOCE abrogation in ORAI1 KO HEK-293. At the opposite, data from Alansary et al. [37] suggested that ORAI3 was responsible for a limited SOCE in HEK-293 cells knocked out for ORAI1 and ORAI2. Our results, where ORAI1 KO HEK-293 cells still presented about 20% SOCE are in line with the latter. Such differences in SOCE might be explained by the strategies used to generate ORAI1 KO. In our study, we used two gRNAs, inducing deletion of the *ORAI1* start codon and thus preventing any ORAI1 protein expression, while others used a single gRNA, leading to the induction of an ORF shift in the middle of the protein coding sequence (*i.e*., end of the first exon or beginning of the second exon) [35,36,37]. While the possibility that the remaining “half-ORAI1” protein could bear semi-functional properties, such as binding partner protein, is highly hypothetical, it represents a potential explanation for the differences observed in Ca^2+^ level entries between our respective ORAI1 KO HEK-293 cells. Alternatively, we hypothesized that the presence of residual SOCE could also result from ORAI2- or/and ORAI3-formed channels because they have been shown to allow limited Ca^2+^ entry when overexpressed [33,38]. For example, it was suggested that ORAI2 could participate in the formation of functional channels. Nonetheless, it was also highlighted that ORAI1 was present in the model studied [12,39,40]. In addition, Alansary et al. [37] suggested that ORAI3 was responsible for a limited SOCE in HEK-293 cells KO for ORAI1 and ORAI2. Thus, we verified that ORAI1 KO HEK-293 cells did not lead to compensatory overexpression of *ORAI2* and *ORAI3* and even showed that these two genes are downregulated. Alternatively, the limited SOCE observed in ORAI1 KO HEK-293 cells could be attributed to channels from the transient receptor potential cation (TRPC) channels, as it has been suggested that they could participate in SOCE [41]. Even though their role in SOCE is dependent on the presence of ORAI1, one can hypothesize that, in the absence of ORAI1, TRPC interact with ORAI2 or ORAI3 to produce limited SOCE. In any case, the definitive exclusion of ORAI2 and ORAI3 as the actors responsible for the limited SOCE observed in our cell lines would require additional experiments such as the investigation of their surface expression or the use of specific inhibitor of these proteins. Unfortunately, the current absence of reliable antibodies targeting ORAI2 and ORAI3, as well as the lack of highly specific inhibitors, prevented us from conducting such experiments.

Furthermore, we studied the effect of ORAI1 deletion and its consequent SOCE impairment on the proliferation and migration of HEK-293 cells. Since the identification of its molecular components, SOCE has been demonstrated to be the main activator of nuclear factor of activated T-cells (NFAT) in immune cells [42,43,44]. In addition to its role in cytokine production, it has also been demonstrated that NFAT regulates key cell-cycle proteins such as cyclins [45]. Interestingly, studies from our group have highlighted the role of ORAI1 in the control of cell proliferation through the utilization of siRNA and a pore mutant of ORAI1 and ORAI3. The studies revealed that ORAI1 is more important than SOCE in the control of HEK-293 cell proliferation [15,16]. Furthermore, our group also demonstrated that ORAI1 silencing is responsible for cell-cycle blockade due to decreased expression of cyclin D1 in prostate cancer cell lines [46]. In addition, Wang et al. [13] confirmed that SOCE alteration, through SOCE inhibitors or ORAI1/STIM1 downregulation, is responsible for cell-cycle arrest in myeloma. Importantly, the recent work of Trebak’s team [35] demonstrated that ORAI1 is required to ensure efficient translocation of NFAT1 and NFAT4 to the nucleus, but the physiological consequences of such translocation were not studied. Similarly, Kar et al. [34] demonstrated, using HEK-293 KO for ORAI1 protein, that the full-length ORAI1 proteins present a specific interaction site with AKAP79 protein. This interaction enables the NFAT1 translocation to the nucleus and the subsequent production of interleukin 5. Nonetheless, the consequences of the alteration of NFAT1 translocation induced by ORAI1 KO on the HEK-293 proliferation were not studied. Given these data, our results indicating that ORAI1 KO does not impact the proliferative rate of HEK-293 might seem surprising. Nonetheless, cancer studies reporting a role for ORAI and SOCE in the control of proliferation denote that they are both dysregulated. Therefore, one could argue that cancer studies do not represent a good model to study the physiological role of SOCE and ORAI. In addition, several other Ca^2+^ channels such as transient receptor potential (TRP) channels or T-type Ca^2+^ channels have important roles in cell proliferation [47]. Similarly, researchers have found that they are deregulated in multiple types of cancer. In a physiological context, studies have highlighted the importance of SOCE for NFATs translocation. Of note, NFATs are involved in the control of several cellular processes such as proliferation, but the consequences of the SOCE-induced NFAT translocation on the proliferation of HEK-293 cells were not studied. Regarding previous data from our laboratory indicating a role of ORAI1 in the control of HEK-293 proliferation [15,16]. The difference with our results, where ORAI1 KO apparently does not affect HEK-293 proliferation, can probably be imputed to the different strategies used in these publications. Indeed, siRNA treatment represents an acute and transitory diminution of expression of ORAI1, while we have induced a permanent and total deletion of this protein. This permanent deletion might lead to the apparition of compensatory mechanisms, enabling a normal proliferation of HEK-293. In addition, HEK-293 cells are cultured in FBS-containing media. Thus, it is conceivable that several external factors control the proliferation of HEK-293 independently of ORAI1 functionality. Overall, because Ca^2+^ plays a crucial role in proliferation, it is probable that alterations in SOCE result in compensatory mechanisms that would allow Ca^2+^ homeostasis to be maintained through the modulation of the expression of other ion channels or receptors. Therefore, ORAI1 KO HEK-293 cells generated in this study could provide a good model for the identification of such modulatory changes by wide transcriptome analysis.

Finally, we studied the consequences of ORAI1 deletion on HEK-293 cells migration properties. Cellular migration is a multistep process involving the creation of a protrusion, the renewal of adhesion spots, and cytoskeletal contraction [48]. Importantly, Ca^2+^—through its action on Ca^2+^-dependent kinases, phosphatases, and protease—has been shown to be involved in all steps of cellular migration [49,50]. In addition, SOCE regulates the migration of several cancer cell types. For example, inhibition of SOCE has been shown to decrease the invasion of melanoma cells [51]. Similarly, the Boyden chamber assay performed with breast cancer cells showed that ORAI1 knockdown decreased their migration level [17]. In a more physiological context, however, it was shown that SOCE suppression did not alter the migration of CD8^+^ T cells [52]. Our data demonstrate a limited role of ORAI1 in the migration process of HEK-293 cells. Specifically, ORAI1 deletion did not affect the adhesive properties of HEK-293 cells. By contrast, in breast tumor cells, ORAI1 knockdown inhibited focal adhesion turnover [17]. However, this was reversed by the transient overexpression of the constitutively active small GTPases Ras and Rac [17]. Of note, these Ca^2+^-dependent proteins are activated by protein kinase C [53]. Interestingly, protein kinase C can be activated independently of SOCE, for example, through G protein-coupled receptor (GPCR) activation. Thus, our result might be explained by the existence of compensatory mechanisms such as modulation of GPCR expression. Our results indicate that ORAI1 KO increases the collective migration speed (wound-healing experiment) but does not affect single-cell migration (Transwell^®^). The difference observed between the two experiments (Transwell^®^ versus wound healing) can be attributed to processes that are specific to a given type of wound-healing experiment (such as cell–cell interactions). Our data suggest that ORAI1 slows the initiation of the collective migration process. Interestingly, in *Xenopus* embryos, the initiation of the leading-edge migration was attributed to Ca^2+^ transients [54]. Indeed, an increasing number of publications indicate that Ca^2+^ oscillations are required for cell migration [18,55,56,57]. Interestingly, Yoast et al. [35] have shown that the absence of ORAI1 does not prevent the appearance of Ca^2+^ oscillations in HEK-293 cells, but it is rather ORAI2 and ORAI3 that are involved in the maintenance of Ca^2+^ oscillations. Thus, one could hypothesize that the increase in the migration initiation speed observed in ORAI1 KO cells is the consequence of a deregulated oscillation mechanism induced by the absence of ORAI1 and the downregulation of ORAI2 and ORAI3. While the initiation of migration is faster in WT HEK-293 cells, our observations show that the overall migration level of ORAI1 KO cells, with significantly reduced SOCE, is equivalent to WT HEK-293 cells. Our data do not eliminate the involvement of other Ca^2+^ signaling pathways such as receptor-operated Ca^2+^ entry and the activation of TRPCs, which have been shown to be involved the migration process [58,59,60,61]. We believe that compensatory mechanisms might have occurred to maintain the physiological migration properties of HEK-293 cells.

## 5. Conclusions

Our study establishes the role of ORAI1 and SOCE in HEK-293 physiology. We have demonstrated that ORAI1 is responsible for the majority of SOCE (~80%) and that proliferation, adhesion and migration rates are barely affected by ORAI1 deletion. Our results indicate that ORAI1 and SOCE are dispensable in the maintenance of HEK-293 physiology. Whether the absence of an effect of ORAI1 deletion is the result of compensatory mechanisms or whether the impact of SOCE in HEK-293 physiology is indeed limited remain to be investigated.

## Figures and Tables

**Figure 1 cells-10-03016-f001:**
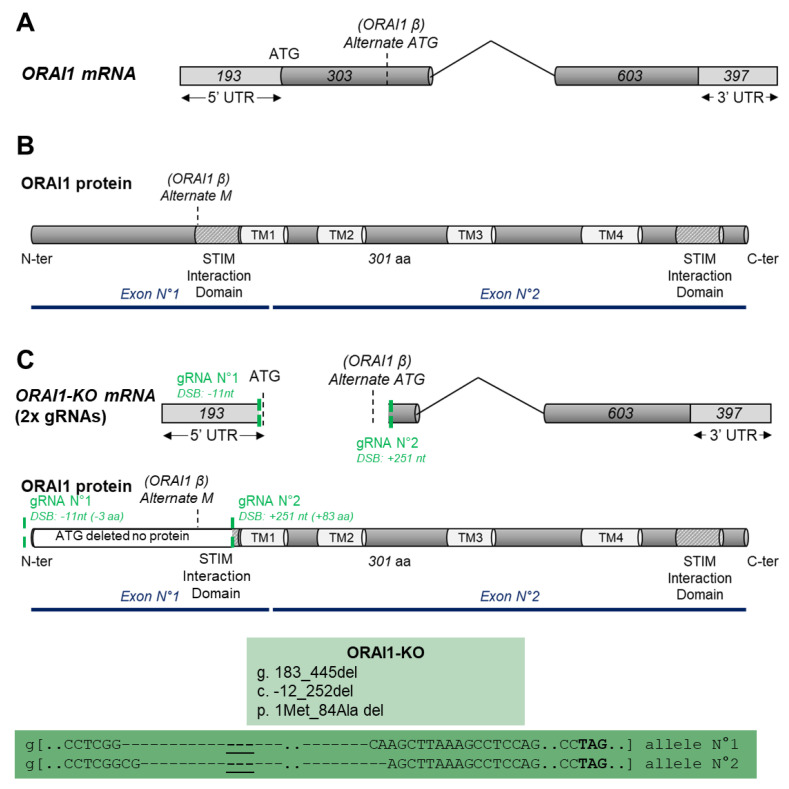
Experimental design of the strategy used to create total knockout of ORAI1 protein (ORAI1 KO). (**A**) Structure of *ORAI1* messenger RNA (mRNA) including the noncoding region (light grey), the coding sequence (dark grey), the original start codon (ATG, full-length ORAI1 origin), and the alternative start codon (ATG, ORAI1β isoform). (**B**) Structure of ORAI1 protein including the transmembrane domain (white boxes) and the stromal interaction molecule (STIM) interaction domain (hatched boxes). (**C**) Guide RNA (gRNA) strategy used to generate complete ORAI1 KO. The green dotted lines represent the localization of the two gRNAs used. Exact coordinates of gRNA double-strand break (DSB) induction sites (relative to the original ORAI1 ATG) are indicated next to the name of the corresponding gRNA (with nt standing for nucleotide and aa standing for amino acid). The light green insert displays the genomic (g), coding sequence (c), and protein (p) coordinates of the induced deletion. The dark green insert presents the sequences surrounding the deletion on genomic DNA level (g) for both alleles (underlined signs indicate the position of the original *ORAI1* start codon).

**Figure 2 cells-10-03016-f002:**
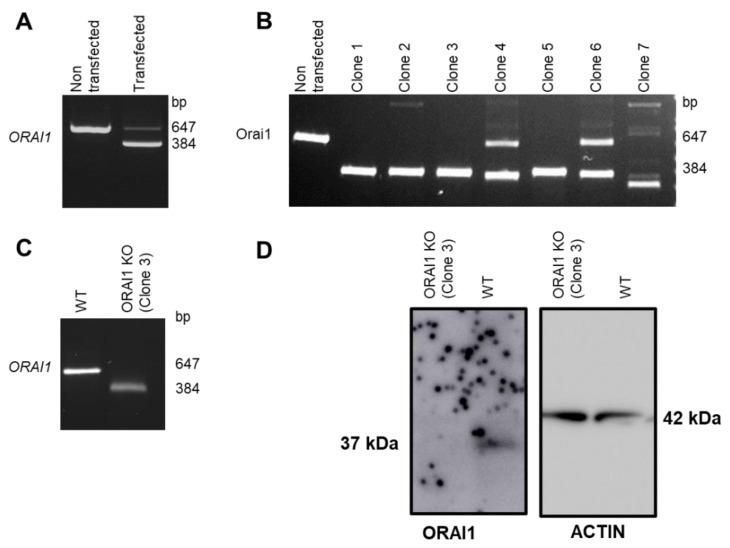
Genomic validation of complete ORAI1 knockout (KO) generation. (**A**) Polymerase chain reaction (PCR) performed on genomic DNA (gDNA) of nonclonal cells corresponding to non-transfected cells and cells transfected with two guide RNAs (gRNAs). Cells transfected with two gRNA present a smaller band (384 bp) indicating the existence of recombination events. (**B**) PCR performed on gDNA of single-cell-derived clones. Clones 1, 3, and 5 presenting a single band around the expected size (384 bp) were selected for further experiments. (**C**) PCR performed on gDNA of wild type (WT) and ORAI1 KO HEK-293 cells. The ORAI1 KO clone presents a lower band indicating successful *ORAI1* gene deletion. (**D**) Western blot analysis of WT and ORAI1 KO HEK-293 cells. WT cells exhibit an ORAI1 band around 37 kDa, while ORAI1 KO cells lack this band.

**Figure 3 cells-10-03016-f003:**
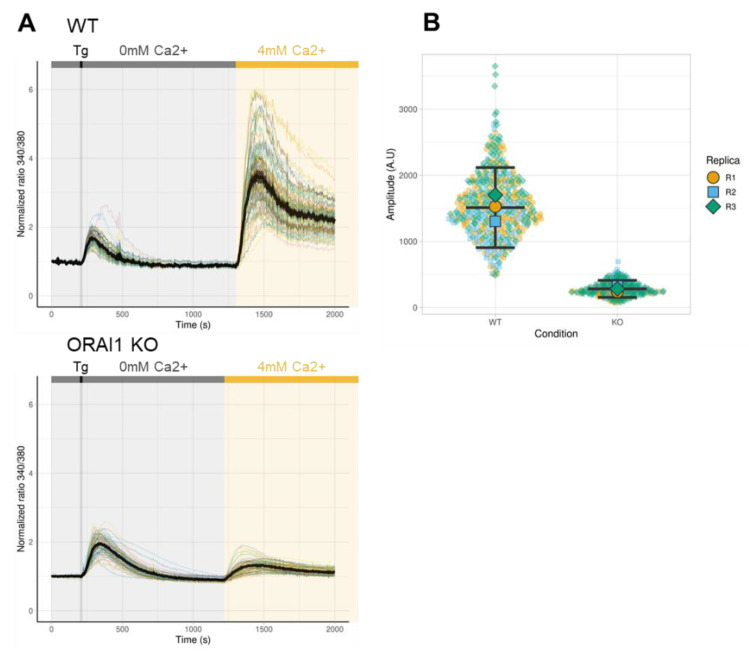
ORAI1 knockout (KO) reduces store-operated calcium entry (SOCE). (**A**) Representative traces of a single Ca^2+^ imaging experiment. For both panels, all recorded cells are represented with different colored lanes, the mean value is presented in black, and the 95% confidence interval (CI) is shown in grey. Each experiment was repeated at least three times. ORAI1 KO cells exhibit a greatly reduced SOCE compared with wild-type (WT) cells. (**B**) The integrated area of the curve following Ca^2+^ addition representing SOCE. Values from each individual cells from three independent experiments are displayed in semitransparent colors (replicate N°1 yellow circles, replicate N°2 blue squares, and replicate N°3 green diamonds). For each experiment, the median values are displayed in plain shapes (replicate N°1 yellow circles, replicate N°2 blue squares, and replicate N°3 green diamonds). Data generated for each experiment were considered nonpaired. Black bars represent the mean and the 95% CI of the three independent experiments. The difference in SOCE between WT HEK-293 cells (mean = 1511 with a 95% CI of 906–2117) compared with ORAI1 KO HEK-293 cells (mean = 282 with a 95% CI of 154–411) was assessed by Welch’s *t*-test and was significant (*p* = 0.007).

**Figure 4 cells-10-03016-f004:**
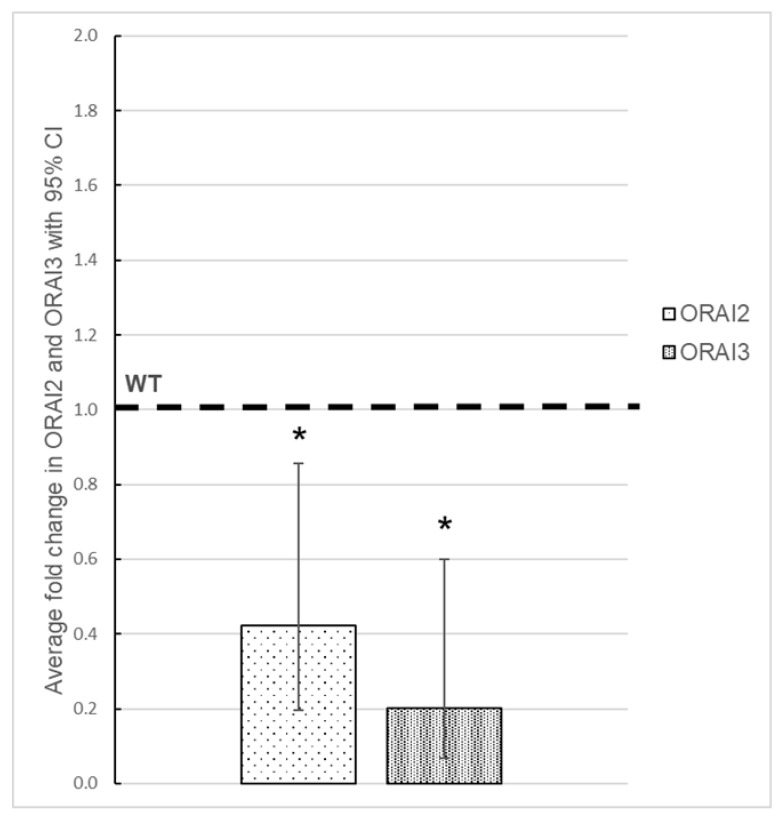
ORAI1 knockout (KO) leads to the downregulation of *ORAI2* and *ORAI3* messenger RNA (mRNA) expression. Dotted boxes represent the mean expression level of *ORAI2* and *ORAI3* mRNA measured by quantitative reverse-transcription polymerase chain reaction (RT-qPCR) from four independent experiments. The levels of *ORAI2* and *ORAI3* mRNA in wild-type (WT) HEK-293 cells were arbitrarily set at 1. In ORAI1 KO cells, *ORAI2* (low-density dots, left box) and *ORAI3* (high-density dots, right box) are downregulated (*ORAI2* relative mean expression level 0.42 with a 95% confidence interval [CI] of 0.19–0.87; *ORAI3* relative mean expression level 0.20 with a 95% CI of 0.07–0.60 compared with WT. The significance of the results was assessed by unpaired *t*-test (* *p* = 0.02 for *ORAI2* and * *p* = 0.01 for *ORAI3*).

**Figure 5 cells-10-03016-f005:**
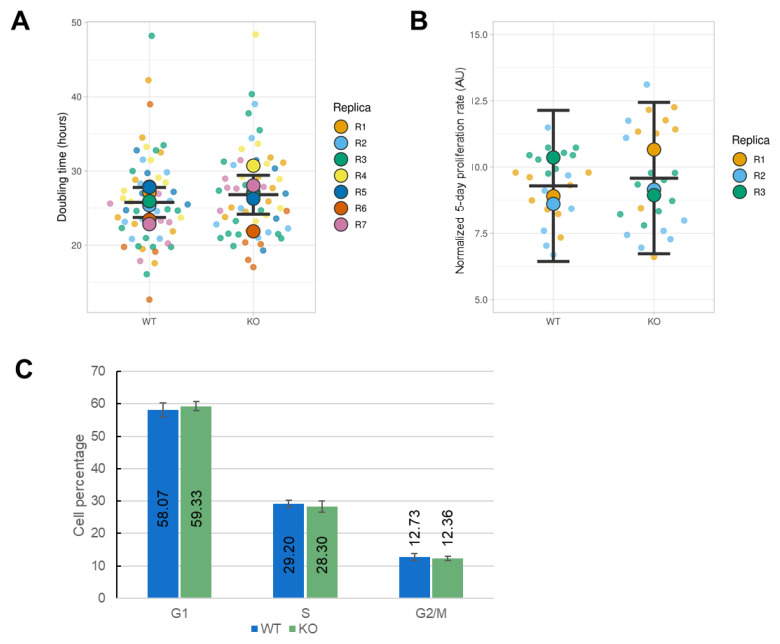
ORAI1 knockout (KO) does not affect proliferation of HEK-293 cells. (**A**) The doubling time of wild type (WT) and ORAI1 KO HEK-293 cells was measured by cell counting over a 5–6 day period. Semitransparent dots represent each doubling time calculated from one experiment (6–14 points per experiment). Plain colored circles display the mean of seven independent experiments. Black lanes represent the median value of each individual experiment and their 95% confidence interval (CI). The difference in the mean doubling time between condition was calculated by using Welch’s *t*-test and was not significant (WT doubling time of 25.77 h with a 95% CI of 23.75–27.79 h; ORAI1 KO doubling time of 26.81 h with a 95% CI of 24.18–29.43 h; difference between the conditions = 1.04 h; Welch’s *t*-test *p* = 0.426). (**B**) The normalized cell proliferation rate assessed by the SRB assay 5 days after seeding WT and ORAI1 KO HEK-293 cells. The semitransparent dots represent single values obtained within each experiment (eight measurements). The median values of three independent experiments are displayed in plain colored circles. The black lanes indicate the mean value of the three individual experiments and their 95% CI (WT = 9.29 with a 95% CI of 6.44–12.14; ORAI1 KO = 9.58 with a 95% CI of 6.73–12.44). The significance of the difference between conditions (−0.29 with a 95% CI of 1.83–2.42) was calculated with Welch’s *t*-test (*p* = 0.721). (**C**) Cell-cycle analysis of the DNA content determined by propidium iodide staining for WT (blue) and ORAI1 KO (green) HEK-293 cells. The histograms represent the percentage of cells in each phase of the cell cycle (exact value indicated within the histogram bars) with error bars indicating the standard deviation. Experiments were repeated six times, and no significant differences between conditions were identified by using Welch’s *t* test.

**Figure 6 cells-10-03016-f006:**
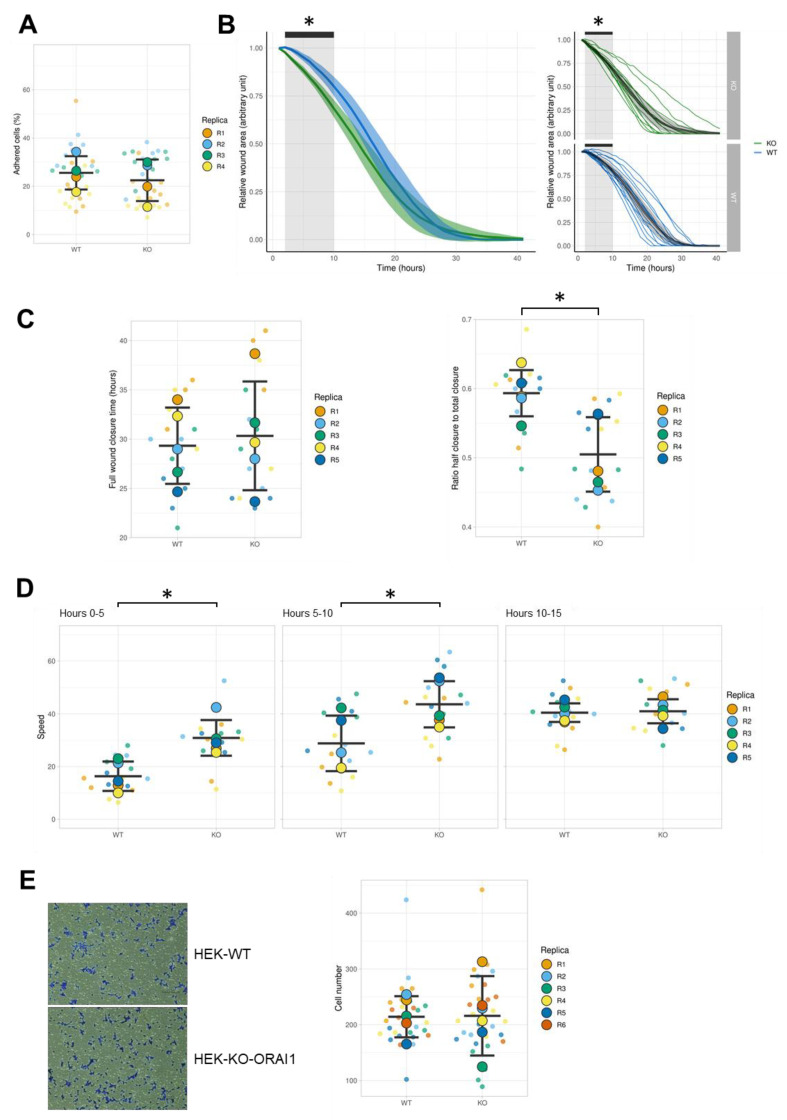
ORAI1 knockout (KO) modifies HEK-293 cell migration properties. (**A**) Comparison of the adhesive properties of wild type (WT) and ORAI1 KO cells. Data points represent the percentage of adhered cells after an incubation for 1 h at 37 °C. Plain circles represent the mean values of individual replicates (R1, R2, R3, and R4); semitransparent circles display individual data points within each replicate. The mean values of the four replicates are indicated as black bars ± standard deviation (CI). The difference in adhesion between WT cells (25.56% ± standard deviation [SD] 6.90%) and KO ORAI1 cells (22.49% ± 8.61%) was assessed with Welch’s *t*-test for nonpaired data and was not significant (*p* = 0.6; mean difference between conditions = −3.07% with a 95% CI of −16.73 to 10.59). (**B**) Middle panel: relative wound area as a function of time (WT in blue, ORAI1 KO in green). The curves represent the means of five independent experiments with their 95% CI (semitransparent colors). Right panels: Relative wound area as a function of time with each repeat presented individually (down panel: WT in blue; top panel: KO in green). Black lines represent the mean value of the experiments with the 95% CI in semitransparent color; the thin colored lanes represent repeats of every single experiment performed. Wound closure area analysis performed for each time point (i.e., every hour) indicates that ORAI1 KO cells migrated significantly faster than WT cells during hours 1–9 of the experiment (* *p* < 0.05 at hours 1–9, assessed based on the wound closure mean value difference between WT and ORAI1 KO cells with Welch’s *t*-test, nonpaired data). (**C**) Left panel: Time of complete wound closure. The mean of each replicate is represented as a plain circle; semitransparent circles represent individual data points for each replicate. The mean of the five replicates is displayed as black bars ± SD. The difference in full closure time between WT cells (29.33 h ± 3.87 h) and KO ORAI1 cells (30.33 h ± 5.51 h) was assessed with Welch’s *t*-test for nonpaired data and was not significant (*p* = 0.75; mean difference between conditions = 1 h with a 95% CI of −6 to +8 h). Right panel: ratio of the half closure time to the total closure time. The mean of each replicate is represented as a plain circle; semitransparent circles represent the individual data points for each replicate. The mean of five replicates is displayed as black bars ± SD. The difference in the ratio between WT cells (0.59 ± 0.03) and ORAI1 KO cells (0.50 ± 0.05) was assessed by Welch’s *t*-test for nonpaired data and was significant (* *p* = 0.018; mean difference between conditions = −0.09 with a 95% CI from −0.16 to −0.02). (**D**) Evolution of leading-edge migration speed over 5 h periods. Hours 0–5 are presented on the left panel, hours 5–10 are presented on the central panel, and hours 10–15 are presented on the right panel. The mean of each replicate is represented as a plain circle; semitransparent circles represent individual data points for each replicate. The mean of five replicates is displayed as black bars ± SD. For hours 1–5, the difference in leading-edge migration speed between WT cells (16.33 ± 5.59 pixel/hours) and ORAI1 KO cells (30.89 ± 6.77 pixel/hours) was assessed by Welch’s *t*-test for nonpaired data and was significant (* *p* = 0.006; mean difference between conditions = 14.56 pixel/hours with a 95% CI of 5.45–23.67). For hours 5–10, the difference in leading-edge migration speed between WT cells (28.81 ± 10.53 pixel/hours) and ORAI1 KO cells (43.65 ± 8.77 pixel/hours) cells was assessed by Welch’s *t*-test for nonpaired data and was significant (* *p* = 0.043; mean difference between conditions = 14.84 pixel/hours with a 95% CI of 0.62–29.06). For hours 10–15, the difference in leading-edge migration speed between WT cells (40.48 ± 3.53 pixel/hours) and ORAI1 KO cells (41 ± 4.56 pixel/hours) cells was assessed by Welch’s *t*-test for nonpaired data and was not significant (*p* = 0.84; mean difference between conditions = 0.52 pixel/hours with a 95% CI of −5.49 to 6.53). (**E**) Transwell^®^ migration assay. Left panel: representative picture of the bottom side of Transwell^®^ insert following 24 h incubation with either WT or ORAI1 KO HEK-293 cells. Right panel, statistical analysis of Transwell^®^ migration experiment: semitransparent circles display the number of cells of each counting performed for each replicate. Plain circles represent the mean value for a single experiment. Black lanes show the median value of the six independent experiments performed with their 95% CI. ORAI1 KO cells (mean cell number = 208 with a 95% CI of 139–278) showed no significant difference in the migrating cell number compared with WT cells (mean cell number = 211 with a 95% CI of 185–238). The significance of the difference between WT and ORAI1 KO cells (-3 cells) was assessed with Welch’s *t*-test (*p* = 0.91).

**Table 1 cells-10-03016-t001:** Oligonucleotides used for cloning, polymerase chain reaction (PCR), and quantitative reverse-transcription PCR (RT-qPCR).

Experiment Type	Name	Sequence 5′→3′
CRISPR gRNA cloning	gRNA1	ATGCATGGAGCACGCCGCCG
	gRNA2	CGCCAAGCTTAAAGCCTCCA
PCR Screening	ORAI1 screening forward	GGCACTTCTTCGACCTCGTC
	ORAI1 screening reverse	CTTGTCACCACCCCAGATCG
RT-qPCR	ORAI1 forward	ATGGTGGCAATGGTGGAG
	ORAI1 reverse	CTGATCATGAGCGCAAACA
	ORAI2 forward	ACCTGGAACTGGTCACCTCT
	ORAI2 reverse	ATGGCCACCATGGCAAAGC
	ORAI3 forward	GGCCAAGCTCAAAGCTTCC
	ORAI3 reverse	CCTGGTGGGTACTCGTGGT
	GAPDH forward	ACCCACTCCTCCACCTTTG
	GAPDH reverse	CTCTTGTGCTCTTGCTGGG
	TBP forward	CTTGACCTAAAGACCATTGCACTTC
	TBP reverse	GTTCTTCACTCTTGGCTCCTGTG

## Data Availability

The data presented in this study are available on request from the corresponding author.

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
