# Peer review of "Impact of SOCE Abolition by ORAI1 Knockout on the Proliferation, Adhesion, and Migration of HEK-293 Cells"

_cells, 2021, doi:10.3390/cells10113016_

Round 1

Reviewer 1 Report

The present study entitled "Impact of SOCE abolition by ORAI1 knockout on proliferation, adhesion, and migration of HEK-293 cells" by Alexandre Bokhobza, Nathalie Ziental, Laurent Allart, Oksana Iamshanova and Fabien Vanden Abeele describes the behavior of Orai1 deficient HEK293 cells with respect to the cellular properties mentioned in the title. This indeed brings an important informative content, as many current and past studies on the molecular mechanisms of STIM and Orai proteins have been performed with Orai knockout cells (HEK293). Of principle, it is important to know how Orai1 KO affects HEK293 cells, which was investigated in the present study. The results show that apart from reduced SOCE, the Orai1 KO does not have any dramatic consequences for the cells. On the one hand, this is surprising because of the important contribution of the SOCE system to intracellular calcium signaling; however, theoretically, compensatory mechanisms may play an important role in these Orai1-deficient cells, as suggested by the authors. In general, the manuscript is well structured and methodologically well prepared. I would suggest the following to improve the quality of the paper:

In the Introduction (lines 33 - 39), citing a recent review would be beneficial. (For example “Mechanism of STIM activation.” Fahrner M, Grabmayr H, Romanin C. Curr Opin Physiol. 2020 Oct;17:74-79. doi: 10.1016/j.cophys.2020.07.006. PMID: 33225118 or “STIM Proteins: An Ever-Expanding Family.” Grabmayr H, Romanin C, Fahrner M. Int J Mol Sci. 2020 Dec 31;22(1):378. doi: 10.3390/ijms22010378. PMID: 33396497

As a non-native English speaker, I am hardly in a position to provide grammatical suggestions for improvement; however, I noticed some errors in the text. Therefore, it is recommended to have a native speaker proofread the paper.

Line 335: the heading is in the wrong format

Line 407: the description of the migration experiments is only visible in textual form. Figure 6 does not appear in the manuscript.

Line 449: The interpretation that an Orai1 protein deleted in the middle still has a functional STIM1 binding domain in the region A73 - R91 is in my view highly speculative. It is absolutely not clear whether a half Orai1 protein will incorporate into the membrane at all, or whether such a protein will be tolerated in the cell at all. Therefore, I would neutralize the statement that a half Orai1 protein can interact with STIM1 and thus Orai2 and Orai3 have insufficient interaction with STIM1.

For the defective Orai1 protein resulting from the knockout (Fig.1), it would be convenient for the reader to know where exactly the deletion is located in the protein.

Reviewer 2 Report

Reviewer’s comments to « Impact of SOCE abolition by ORAI1 knockout on proliferation, adhesion, and migration of HEK-293 cells » by Bokhobza et al.

In this work Authors created ORAI1 knock-out HEK-293 cells using CRISPR/Cas9 technology and an original targeting approach, and investigated various fundamental cell functions such as proliferation and migration. Capacitative calcium influx/store operated calcium entry and corresponding ORAI1 have been shown to be important players in the regulation of various cellular processes including proliferation and migration, and this is currently an active research field for, example, in cancerology. Some uncertainly lingers, however, because in the past experiments relied mainly on knock-down that may lead to partial effects, and on pharmacological inhibitors, which sometimes require complex interpretations due to selectivity-related issues. In addition, previous knock-out experiments may also be associated, as pointed out by the Authors, with the existence of remaining ORAI1 fragments that may possess ill-defined residual activities.

The Manuscript contains an abundant set of relatively minor errors of English grammar; these absolutely need to be corrected before publication, as they considerably disturb reading. This notwithstanding, this work is highly commendable and timely, constitutes an important advance in the field, will undoubtedly prompt further studies and clarify issues, and will interest a wide range of readers.

Comments:

Lane 18 : « …the absence of SOCE does not deeply impacts (sic) HEK-293 cells… » : « does not deeply impact »

Please replace « Ca2+ » by « Ca2+ » in the text.

Lane 565 : Please check Ref. 8.

Lane 54 : « SOCE did not control proliferation of renal metastatic cells » It is not clear, whether Authors refer to renal metastases of carcinomas or to metastatic renal cell carcinoma cells.

Lane 60 : « herein » or « thereof » ?

Lane 82 :  “CO2” : “CO2”; idem “H2O” in lane 86. Idem l. 150-156 etc.

Quid mycoplasma status of the cell line ?

Lane 88 “After filtration through 0.22μm syringe…” is a somewhat sloppy description; probably a 0.22µm pore-size filter and a syringe were used.

Lane 94 : Authors state that transfection reagent mix was prepared with FCS-free medium. Please specify whether cells were transfected in the absence or the presence of FCS, and whether washing steps with FCS-free medium were done prior transfection. The detailed, precise and exact description of the transfection and CRISPR/Cas9 experiments is important for reproducibility.

Lane 119 : “Following cool down of the tubes a last vortex step was performed…” : “Following cool-down of the tubes, a last vortex step was performed…”

Lane 124 : “PCR on gDNA were performed…” : “…was performed” ?

Lane 127 :  “Cloning of PCR product for sequencing were performed…”  :  “…was performed…” ? (idem l. 131 and 132)

Lane 159 : “…with Poly-L-Lysine” : “…with poly-“

Lane 160 : “diluted in 2 Ca2+ HBSS solution…” :  2 mM? (idem l. 165)

Lanes 150-157 : Considering that EGTA is also present, maybe actual free calcium ion concentrations could be given (as measured using a calcium selective electrode or calculated). It is not entirely clear, whether old (EGTA-containing) medium was replaced by aspiration before addition of new (calcium-containing) medium, or just added upon.

Lane 177 : “ 1x105 ” : “ 1x105 ”, idem l. 186 etc.

Lane 181 : “was read using flow cytometer CyAn™ ADP Analyser FACScan flow cytometer…” : “using a CyAN… flow cytometer…” ?

Lanes 172 and 184 : “excel” or “Excel” ?

Lanes 190-191 : Please present the formula in a more intelligible way.

Lane 195 : “5 x103 cells per well were plated in 96 well-plates. 5 days after seeding, cells were fixed with addition of 50% Trichloroacetic acid” : one imagines that medium was withdrawn and cells were washed before addition of TCA.

Lane 198 : “ ((Sigma-Aldrich)” : “(Sigma-Aldrich)” ?

Lanes 204-205 : “96 well-plates were coated with fibronectin by the transient addition of fibronectin solution (10 µg/ml) in each well before air dry the residual liquid for 30 min under the hoot (sic).” : “under the hood” ? And also, please specify the time the fibronectin solution spent in the wells, and whether there was then an aspiration step.

Lane 212: “webapp, for…” : “webapp; for…”

Lane 227 : “Time series graph were generated…” : “Time series graphs were generated…” ?

Lane 233: “assay were” : “assays were” ; “with a size pore of” : “with a pore size of”

Lane 239 : “1% crystal violet in 20% methanol solution” : “1% crystal violet solution in 20% methanol”

Lane 240 : “number of cells migrated” : number of cells that migrated “

Lane 246 : “SuperPlotsOfDarta” : “Data” ?

Lane 252 : “and within technical replicate” : “replicates” ?

Lane 255 : “Errors bars” : “Error bars” ?

Lanes 260-262 : “Since the compared cell lines were genetically modified and thus evolved differently following genome editing experiments, the data were considered unpaired, thus statistical test used was Welch’s t-test, except if stated otherwise.” This requires a clearer explanation.

Lanes 276-281 : “To facilitate and ensure the generation of a total ORAI1 KO, we decided to induce two DSB: one located above the classic ORAI1 start codon (i.e: nucleotides 194-196 from NCBI: NM_032790.3), and the second located downstream of the alternative -ORAI1 β start codon (i.e: nucleotides 404-406 from NCBI: NM_032790.3). Thanks to this approach, we hypothesize that we would induce a genomic deletion that includes ORAI1 start codon of the two identified isoforms and would lead to the generation of complete ORAI1 KO.” :  The use of the term “above” is somewhat unfortunate here, because “above” in this context may be understood by the reader either as “upstream of the classic ORAI1 start codon”, or as “on top of/located exactly at” the start codon. Location of the two DSBs should be stated unambiguously in the text. Fig.1 in its present form is not very helpful to clarify this issue either. In addition, the mechanism of the desired deletion of the in-between region should also be discussed briefly.

Also, please discuss whether ORAI2 and 3 are/are not targeted by the guide RNAs used based on the sequences.

Lane 310: “underlined signs indicates…” : “indicate” ?

Lane 315 : “indicating on the…” : “indicating the…” ?

Fig.1, Panel A : Because a minor band can be seen in the transfected cells at 647 bp, it would be nice to point out in the legend that in this experiment the cells were not clonal.

Also, considering that ORAI2 and 3 are discussed in the paper, the Reviewer wonders whether ORAI2 and ORAI3 western blots could be performed and shown.

Fig.1, Panel D : If possible it would be nice to show a better, and full-lenght western blot image in order to illustrate the complete absence of a specific band in KO cells. Where is ORAI1 b in the WT cells in the westerns ? Which KO clone is depicted in this Panel ?

Lane 324: “Thapsigargin (TG) is irreversible inhibitor of sarco/endoplasmic reticulum Ca2+-ATPase”: “is an irreversible…” ?

It is stated in the Legend of Fig.1 that several KO clones have been obtained (clones 1, 2, 3 and 5). At the same time in Fig.2 the ~80% abolition of SOCE in KO cells is shown. Was this done on a single clone (out of the four shown)?

Lanes 357-358  : “Taken together, these results suggest that that the residual SOCE observed in ORAI1 KO cells is unlikely provided by overexpression of ORAI2 and ORAI3.” : maybe it would be appropriate for clarity to state here, what is the relevance of an eventual overexpression of ORAI2 and 3 ? Couldn’t the observed residual SOCE be attributed to baseline ORAI2 and 3 expression ? This needs to be discussed here also in the context of Refs. 28, 30 and 31. If baseline expression of ORAI1 and 3 (unlike overexpression) is considered irrelevant in the context of compensatory SOCE, this should be discussed here, and not only later in the Discussion (see also lanes 454-458). In brief, the issue of the eventual role of overexpression versus baseline expression of ORAI2 and 3 should be addressed in a more straightforward manner.

Lane 362 : “expression.Histogram…” : “expression. Histogram…”

Lane 364 : “was arbitrary setup at 1.” : “was arbitrary set at 1.” ?

Fig. 4 : Please state to what do the two dotted boxes correspond to mathematically.

Lane 387 : “growth.(A)…” : “growth. (A)”

Lane 391 :  “The difference in the mean doubling between condition…” : ?

Lane 396 : “Orai1” : “ORAI1” ? “singles values” : “ single values » ?

Lane 387 : “The median values of 3 independent experiment are displayed…” : “experiments”

Lane 405 : “and no significant difference between conditions were identified…” : “difference … was identified…” (or: “…differences … were identified…”)

Lane 416 : “until the complete wound closure” : “until complete wound closure” or “until the complete closure of the wound”

Lane 420 : “in the collective speed migration” : “speed of migration” ?

Lane 425 : “between the two migration front” : “fronts” ?

Lane 431 : “our data of wound healing experiment indicates that…” : “experiments…” ? (and “presents a delay…)

Lanes 432-433 “To elucidate whether the effect observed was restricted to collective migration or whether it was affecting the global migration process…” It is not entirely clear, what does “global” mean in this context.

Lane 454 : “despite the complete ORAI1 depletion…” : “despite complete ORAI1 depletion” or “despite the complete depletion of ORAI1”

Lane 456 : “Additionally, few publications showed that ORAI2 could form functional channel” please reformulate for clarity. (“only a few” ?)

Lane 480 : “pointing onto” ?

Lane 481 : “Since ORAI1 and SOCE are deregulated in cancer, it hampers the identification of…” please reformulate.

Lane 488 : “since Ca2+ plays crucial role…” : “…a crucial role…” ?

Lane 497 : “inhibition of SOCE have been…” : “has been…”

Lane 501: “[44].Our…” : “[44]. Our…”

Lane 506 : “activated by the protein kinase C” : “activated by protein kinase C” ?

Lane 510 : “The results of wound-healing experiments indicates that…” : “results … indicate…”

Lane 514 : “specific to wound healing experiment…” : “specific to a given type of a wound healing experiment” ?

Lane 518 : “Interestingly, Yoast et al., have shown that…” : “Interestingly, Yoast et al. have shown that…”

Lanes 522-523 : “is the consequence of deregulated oscillation mechanism…” : “…of a deregulated oscillation mechanism…” or “of deregulated oscillation mechanisms…”

Lane 524 : “our observations shows that…” : “our observations show…” or “our observation shows…”

Lanes 526-527 : “Our data does not eliminate involvement of other Ca2+ signaling pathways such as receptor-operated Ca2+ entry and TRPCs activation which have been shown to be involved in migration process” : “Our data does not eliminate the involvement of other Ca2+ signaling pathways such as receptor-operated Ca2+ entry and the activation of TRPCs, which have been shown to be involved in the migration process”

Lane 534 : “…deletion., Our…” : “…deletion. Our…”

Lanes 536-537 : “Whether the absence of effect of the ORAI1 deletion is the results of compensatory mechanisms or whether of the impact of SOCE in HEK-293 physiology is indeed limited remains to be investigated.” : “Whether the absence of effect of ORAI1 deletion is the result of compensatory mechanisms or whether the impact of SOCE in HEK-293 physiology is indeed limited, remains to be investigated.”

Reviewer 3 Report

Bokhobza et al. in their manuscript cells-1431068 present their findings on the role of Orai1 in HEK293 proliferation, adhesion and migration. The role of Orai1 in SOCE has been well established by numerous siRNA studies and in addition Orai1 knockout by CRISPR/Cas9 and its impact on SOCE has been reported in several studies for example Cai et al., JBC, 2016 and Alansary et al., JCS, 2020, which the authors did not cite. Therefore, the findings presented by the authors in this context are trivial. Further, the authors do not discuss the work of El Boustany et al., Cell Calcium, 2010 in context of the role of Orai1 in the proliferation of HEK-293 cells. In lines 55-57, the authors state that “their lab has shown that Orai proteins were important for the growth of HEK293 but SOCE was dispensable for their proliferation”. While Borowiec et al., BBA, 2014 claimed that HEK293 proliferation depends on Orai1, Bokhobza et al. in their manuscript present contrasting findings. How do the authors reconcile this? This should be discussed. The data referred in text on the adhesion and migration does not suggest any striking differences which makes the overall impact of the story “weak”. Also, please note that the manuscript did not contain figure 6, which makes it difficult for the reviewer to understand the data. Please see additional comments below:

- Due to the presence of small amount of SOCE in HEK293 Orai1 KO cells, the authors rationalize to test for any compensation by Orai2 and Orai3. Therefore, Bokhobza et al. test the mRNA expression levels of Orai2 and Orai3. The data they present in figure 4 has very high variability and the authors should have further worked on it to improve this by increasing the sample size. Nevertheless, because the authors did not observe any upregulation of other Orai isoforms in Orai1 KO HEK293 cells, the authors argue that the left over SOCE is not due to upregulation of Orai2 or Orai3. If the authors want to propose this rationale, they should rather spend some effort on investigating the surface expression levels of Orai2 and Orai3 in HEK293 Orai1 KO cells (see Alansary et al., BBA, 2015). In addition, they should also try Orai specific inhibitor GSK-7975A on the leftover SOCE in Orai1 KO HEK293 cells to test if that is wiped out.

- Line 350: The authors are suggested to also cite Kar et al., PNAS, 2021 and Yoast et al., Nat. Comm., 2020 wherein both studies showed that Orai2 and Orai3 can generate SOCE in HEK293 Orai triple knockout cells.

- Lines 476-478: In addition to the currently cited work (ref #30), the authors are advised to cite the work of Kar et al. 2021. To provide the authors a brief overview, Kar et al. 2021 performed in-depth analysis of the interaction site of AKAP79 on Orai1, a site which is lacking in Orai2, Orai3 as well as the shorter variant of Orai1 (Orai1_beta). They demonstrated how Orai1_alpha-AKAP79 interaction facilitates rapid NFAT1 nuclear translocation upon local Ca2+ entry through Orai1 channels. Kar et al. could prevent this interaction by a peptide derived from the N-terminus of Orai1 which could block NFAT1 nuclear translocation without affecting SOCE and using the same peptide they could prevent IL-5 cytokine production through NFAT1 pathway. In Orai1 knockdown HEK293 cells, Kar et al. 2021 showed that Orai1_beta, Orai2 and Orai3 are inefficient in driving NFAT1 nuclear translocation and this finding for Orai3 was confirmed using CRISPR/Cas9 mediated Orai knockout HEK293 cells. In addition to this and in the same context, please consider to cite the earlier findings of Feske et al., Nature, 2006 that highlighted the Orai1-NFAT1 connection in immunodeficiency (SCID).

- Line 480-481: Herein the authors should make a case why it is important to address the role of Orai1 in cell proliferation specifically in the HEK-293 cells?

- Line 414: ORAI KO should be ORAI1 KO

Round 2

Reviewer 2 Report

The Reviewer would like to suggest the following minor modifications:

1.

Point 2: Lane 54 : « SOCE did not control proliferation of renal metastatic cells » It is not clear, whether Authors refer to renal metastases of carcinomas or to metastatic renal cell carcinoma cells.

Response 2: We have clarified as follow [please see the lanes 62-64 from the revised manuscript version: “On the other hand, pharmacological and genetic inhibition of SOCE has no effect on the proliferation of cells originating from renal metastases.”[14].

[14] Dragoni, S.; Turin, I.; Laforenza, U.; Potenza, D.M.; Bottino, C.; Glasnov, T.N.; Prestia, M.; Ferulli, F.; Saitta, A.; Mosca, A.; et al. 692 Store-Operated Ca 2+ Entry Does Not Control Proliferation in Primary Cultures of Human Metastatic Renal Cellular Carcinoma. 693 Biomed Res. Int. 2014, 2014, 1–19, doi:10.1155/2014/739494.

Question: It is still unclear what these cells are : “cells originating from renal metastases” means malignant cells that have formed metastases in the kidney : “Renal metastasis” is a metastasis in the kidney (presumably of non-renal origin; although a renal metastasis of a primary, for example contralateral, renal cell carcinoma is also a possibility.) On the other hand, “primary cultures of human metastatic renal cellular carcinoma” (as stated in the title of the given Reference) refers to cells cultured from distant metastases of a primary renal cell carcinoma. Please clarify. In Ref. 14 it is stated that : “Tumour samples were collected from 4 patients affected by metastatic renal carcinoma (mRCC) who have undergone surgical intervention or biopsy to remove metastasis".

2.

Point 3: Quid mycoplasma status of the cell line.

Response 3: We have added in the material and method section that the cell lines were checked for mycoplasma contamination through Hoechst staining [please see the lanes 103-104 from the revised manuscript version:” All cells used in this study were validated as being mycoplasma free by using Hoechst staining.”

In the revised version of the Manuscript received by the Reviewer, lanes 103-104 are as follows:

“…the T7 Endonuclease I assay (NEB) according to the manufacturer’s instructions. The 103 choice of gRNAs was dictated by their efficiency and the number of predicted off-target…”

Q: The mentioned text is in lanes 83-84. This type of problem applies to all other modifications as well.

3.

Lane 179 : “(calculated free final Ca2+ concentration = 3.95 mM)”

Q: please include software name and link, if available.

4.

Response 7: The formula is now written as follow [Please see the lanes 249, 250 from the revised manuscript version:

Doubling time (hours) = [Duration (hours) × log 2] / [log (Cell Number (Day x)) – Log (Cell number (Day x – 1))].

Please consider that the initial formula was written using the Equation option from MS Word but was apparently modified during the editor formatting. As a consequence, we did not use the Equation option to re-write it.

Q: Unfortunately, the Reviewer finds this new way of expressing the formula still a little confusing. Probably a correctly formulated/formatted equation, integrated in the text as an image file would be the solution.

“Duration” of what ?

5.

Point 8: Lane 195 : “5 x103 cells per well were plated in 96 well-plates. 5 days after seeding, cells were fixed with addition of 50% Trichloroacetic acid” : one imagines that medium was withdrawn and cells were washed before addition of TCA.

Response 8: We are now indicating that cell medium was removed before addition of TCA [ Please see the lane 258 from the revised manuscript version: “the cell medium was discarded”]

Q: TCA will precipitate proteins (FCS) present in the culture medium eventually remaining in the wells; there is quite some capillarity at the bottom of a well of a 96 well plate. Therefore the Reviewer would like to ask again, whether a washing step (with a protein-free solution such as, for example, PBS) was included.

6.

Point 9: Lanes 204-205 : “96 well-plates were coated with fibronectin by the transient addition of fibronectin solution (10 µg/ml) in each well before air dry the residual liquid for 30 min under the hoot (sic).” : “under the hood” ? And also, please specify the time the fibronectin solution spent in the wells, and whether there was then an aspiration step.

Response 9: The fibronectin coating procedure has been modified to provide more details [Please see the lanes 270-273 from the revised manuscript version: “Adhesion assays were performed in fibronectin pre-coated 96-well plates. Briefly, 20 µL of a fibronectin solution (10 µg/mL) was added to each well for 5 minutes at room temperature. Following aspiration of fibronectin solution, plates were left uncovered under the hood for 30 minutes to air dry the residual solution”].

Q: Fibronectin was dissolved in what ? This needs to be stated for reproducibility, as other components (such as, for example, serum proteins or other polymers) in such a solution may interfere with the adhesion of fibronectin to plastic. Presumably a calcium-free PBS solution was used.

7.

Q: Was the analysis of potential off-target sites regarding the CRISPR/Cas9 experiments based on a normal human genome or a HEK293 genome ?

8.

Do the Authors believe that the ORAI1 western blot in Fig 2 is acceptable in its present form ? There are many parasitic black dots in this image, quite unusual for a wb. How many times was this western blot performed ? Is this the best image obtained by the Authors ? Is ORAI1 western blotting a particularly difficult undertaking ? In order to show specificity, it would be nice to show a full-length blot (and molecular weight markers).

9.

Point 21: Lane 481 : “Since ORAI1 and SOCE are deregulated in cancer, it hampers the identification of…” please reformulate

Response 21: We have reformulated this sentence [Please see the lanes 698-701 from the revised manuscript version: “Nonetheless, cancer studies reporting a role for ORAI and SOCE in the control of proliferation denote that they are both dysregulated. Therefore, one could argue that cancer studies does not represent a good model to study the physiological role of SOCE and ORAI.”]

“…cancer studies do not represent…”

10.

As stated by the Authors, a single new RT-qPCR experiment modified the conclusion shown in Fig. 4 (statistically significant ORAI2 down-regulation). The absence of an increase of ORAI2 and 3 mRNA expression is convincing, and thus this does not modify the conclusions of the Manuscript. However, considering the impact of the single additional ORAI2 RT-qPCR experiment on the statistical significance of ORAI2 mRNA levels (unchanged or decreased), the Reviewer has the impression that doing several additional RT-qPCR experiments would be warranted, eventually.

11.

Please note that the name of the cell line at the ATCC is "HEK-293".
